# Oxidative killing of encapsulated and nonencapsulated *Streptococcus pneumoniae* by lactoperoxidase-generated hypothiocyanite

Aaron D. Gingerich[1], Fayhaa Doja[1], Rachel Thomason[1], Eszter Tóth[1], Jessica L. Bradshaw[2], Martin V. Douglass[1], Larry S. McDaniel[2], Balázs Rada[1] *

1 Department of Infectious Diseases, College of Veterinary Medicine, The University of Georgia, Athens, Georgia, United States of America, 2 Department of Microbiology and Immunology, University of Mississippi Medical Center, Jackson, Mississippi, United States of America

* radab@uga.edu

**Data Availability Statement:** All relevant data are within the manuscript and the raw data are

## Abstract

*Streptococcus pneumoniae* (Pneumococcus) infections affect millions of people worldwide, cause serious mortality and represent a major economic burden. Despite recent successes due to pneumococcal vaccination and antibiotic use, Pneumococcus remains a significant medical problem. Airway epithelial cells, the primary responders to pneumococcal infection, orchestrate an extracellular antimicrobial system consisting of lactoperoxidase (LPO), thiocyanate anion and hydrogen peroxide ($H_2O_2$). LPO oxidizes thiocyanate using $H_2O_2$ into the final product hypothiocyanite that has antimicrobial effects against a wide range of microorganisms. However, hypothiocyanite's effect on Pneumococcus has never been studied. Our aim was to determine whether hypothiocyanite can kill *S. pneumoniae*. Bactericidal activity was measured in a cell-free *in vitro* system by determining the number of surviving pneumococci via colony forming units on agar plates, while bacteriostatic activity was assessed by measuring optical density of bacteria in liquid cultures. Our results indicate that hypothiocyanite generated by LPO exerted robust killing of both encapsulated and nonencapsulated pneumococcal strains. Killing of *S. pneumoniae* by a commercially available hypothiocyanite-generating product was even more pronounced than that achieved with laboratory reagents. Catalase, an $H_2O_2$ scavenger, inhibited killing of pneumococcal by hypothiocyanite under all circumstances. Furthermore, the presence of the bacterial capsule or lytA-dependent autolysis had no effect on hypothiocyanite-mediated killing of pneumococci. On the contrary, a pneumococcal mutant deficient in pyruvate oxidase (main bacterial $H_2O_2$ source) had enhanced susceptibility to hypothiocyanite compared to its wild-type strain. Overall, results shown here indicate that numerous pneumococcal strains are susceptible to LPO-generated hypothiocyanite.

uploaded to Dryad: https://doi.org/10.5061/dryad.547d7wm5t.

**Funding:** This study was supported by the following: 1. BR. R21AI124189-01A1 National Institutes of Health, National Institute of Allergy and Infectious Diseases R21AI124189-01A1 URL: https://www.niaid.nih.gov 2. BR, startup funds provided by the University of Georgia Office of Vice President for Research URL: https://research.uga.edu; 3. BR. 1 R21 AI147097-01A1 National Institutes of Health, National Institute of Allergy and Infectious Diseases URL: https://www.niaid.nih.gov The funders had no role in study design, data collection and analysis, decision to publish, or preparation of the manuscript.

**Competing interests:** The authors have declared that no competing interests exist.

## Introduction

*Streptococcus pneumoniae (Spn)* is a leading cause of bacterial infections such as otitis media, pneumonia, septicemia and meningitis [1, 2]. Colonization can occur at any point in a person's life but occurs most commonly in young children where *Spn* prevalence reaches over 50% in hosts 2–3 years old [3]. Worldwide, *Spn* is a major cause of infant mortality with 1.2 million deaths reported every year [2, 4]. Current pneumococcal vaccines target the capsular polysaccharide of *Spn*, but these vaccines only provide serotype-specific protection against less than one third of circulating serotypes [5]. *Spn* infections can also be controlled with antibiotics, but widespread antibiotic use has led to accelerated antibiotic resistance in *Spn* [6]. These challenges have led to the need for novel therapeutics and a better understanding of *Spn* interactions with the host.

The airway epithelium represents one of the largest physical and immune barriers against airborne microbes such as *Spn* [7]. Lactoperoxidase (LPO) is a heme peroxidase found in the airway surface liquid (ASL) where it performs its antimicrobial activity [8]. The LPO-based antimicrobial system requires two other components to function properly. First, LPO needs a source of $H_2O_2$ to catalyze the reaction. In the human airways, $H_2O_2$ is mainly supplied by the NADPH oxidase Dual oxidase 1, Duox1 [9–11]. LPO then uses $H_2O_2$ to oxidize the pseudohalide thiocyanate ($SCN^-$) which is abundantly present in the ASL into the antimicrobial ion hypothiocyanite ($OSCN^-$) [9, 12]. The LPO-based system has previously been shown to be an effective *in vitro* neutralizer of a wide variety of viruses [13–15] and bacteria [8, 16]. Interestingly, even though *Spn* represents an enormous health burden, the effectiveness of the LPO-based system against *Spn* has not been tested so far.

Due to the relevance of *Spn* in public health combined with the emergence of antibiotic resistance, the LPO-based system could provide valuable insight and a possible new therapeutic option for management of *Spn* infections. We hypothesized that the LPO-based system is effective at killing *Spn in vitro*. We found that both encapsulated and nonencapsulated pneumococci were susceptible to $OSCN^-$ mediated killing in a cell-free experimental system.

## Materials and methods

### Bacteria

*Spn* strains EF3030 (encapsulated serotype 19F) [17], EF3030 Δ*cap* (isogenic, nonencapsulated mutant strain) [18], MNZ41 (nonencapsulated) [19], TIGR4 (encapsulated serotype 4) [20], TIGR4Δ*spxB* (isogenic mutant deficient in pyruvate oxidase) [21] and TIGR Δ*cap* (isogenic mutant deficient in capsule formation generated by the same strategy as EF303 Δ*cap*) [18], D39 (encapsulated serotype 2) and its isogenic, capsule-free mutant, D39 Δ*cap* [22] were inoculated on sheep blood agar plates (BAP) and incubated at 37°C in 5% $CO_2$. After incubation, bacteria were collected and harvested by centrifugation at 10,000g for 5 minutes, washed twice with Hank's balanced salt solution (HBSS), and suspended in HBSS. Bacterial density was then determined by measuring optical density (OD) at 600 nm. The bacterial density was set to 0.6, which is representative of $10^9$ CFU/mL *Spn* that was confirmed by performing serial dilutions, plating bacteria on BAP and counting colonies. Bacteria were prepared this way for both, bacterial killing and bacteriostatic measurements. The identities of the *Spn* strains were confirmed by 16S rRNA Gene Sequencing (Genewiz, South Plainfield, NJ, USA). Optochin-sensitivity of the *Spn* strains used was also confirmed for each experiment using BD BBL™ Taxo™ P Discs (Fisher Scientific, Pittsburgh, PA, USA).

### Bacterial killing measured by colony counting

Components of the LPO-based antibacterial system were used as described previously [8]. Briefly, the following concentrations were used: 6.5 μg/ml LPO, 400 μM $SCN^-$, 5 mM glucose

and 0.1 U/mL glucose oxidase. The reaction volume was set to 120 μL. Catalase (700 U/mL) was also used when indicated to inhibit the system by scavenging $H_2O_2$. The components were assembled in a sterile 96-well microplate in triplicates with the bacteria being added last at a maximal concentration of $5x10^5$ CFU/ml. The plates were then placed in a 37°C incubator with 5% $CO_2$. After 6 hours of incubation, 40 μL was spread onto BAP in triplicate and incubated at 37°C with 5% $CO_2$. After 24 hours, the colonies were counted and CFU/mL was determined. Agar plates exposed to only the assay medium without *Spn* were always used to ensure that no potential contaminants were detected. A time 0 condition was also counted to make sure that bacterial death was due to $OSCN^-$ and not related to an unknown variable, and that no significant changes in bacterial numbers were observed in samples containing only bacteria during the duration of the experiments. All the reagents were ordered from Sigma-Aldrich (St. Louis, MO, USA) unless stated otherwise.

## Bacteriostatic activity measured by a microplate-based growth assay

The bacteriostatic activity of $OSCN^-$ was measured by a microplate-based assay described previously [23]. Briefly, the components (mentioned in the cell-free assay) were assembled in a sterile 96-well plate with the bacteria being added last. Bacterial growth was measured in a microplate spectrophotometer [Eon (BioTek Instruments Inc., Winooski, VT, USA) or Varioskan Flash (Thermo Scientific, Rochester, NY)] on the basis of following increases in OD as a measure of bacterial density. This method enables fast and very reproducible measurement of bacterial growth [23]. *Spn* strains were grown at 37°C for 14 hours, and OD at 600 nm was measured every 3 minutes. Each sample was run in triplicate. The time required for the positive control (*Spn* alone) to reach an OD of 0.4 (exponential growth phase) was used as the reference point for all other conditions, and OD values of other samples were compared to this. All the *Spn* strains used in this work were tested individually for their suitability for this method.

## The commercial 1$^{st}$ line™ immune support product

*1$^{st}$ line*™ is an over-the-counter product that is marketed as an immune supplement (distributed by Profound Products). This product uses a proprietary technology to keep $OSCN^-$ stable for a longer period of time allowing for a better antimicrobial effect. To our knowledge, this product is the only commercially available product producing $OSCN^-$. We tested this product in conjunction with our previously described cell-free system. Briefly, 0.1 g of LPO was reconstituted in 25 mL of HBSS. 750 μL of $H_2O_2$ solution was added to a 15 mL conical tube followed by the addition of 700 U/mL of catalase. The solution was incubated for 10 minutes to allow catalase to scavenge all $H_2O_2$ present. 12.5 mL of LPO solution was then added to each tube and mixed. Following this step, 750 μL of $SCN^-$ was also added to each sample and mixed thoroughly. Finally, 750 μL poly aluminum chloride was administered to each solution and mixed well. Samples were then incubated for 30 minutes at room temperature allowing the generation of $OSCN^-$. By the end of this incubation time, the solution separates into two distinct phases. The top, clear phase containing $OSCN^-$ was used for experiments while the pelleted precipitate was discarded.

## Bacterial $H_2O_2$ production

Generation of $H_2O_2$ in bacterial suspension was measured by the ROS-Glo™ luminescence kit following the manufacturer's instructions (Promega Corporation, Madison, WI, USA). This sensitive assay enables specific and direct detection of low amounts of $H_2O_2$. TIGR4 wild-type or Δ*spxB* bacteria ($5x10^6$/ml) suspended in HBSS buffer were incubated at 37 °C for 30

minutes, followed by centrifugation to collect supernatants for analysis of $H_2O_2$ production. Ros-GLo[TM] reagent was added to bacterium-free supernatants and luminescence was read using a Varioskan Flash microplate luminometer (Thermo Scientific, Rochester, NY). The assay was run in triplicates. Results are expressed as relative luminescence units (RLU).

### Quantitation of OSCN[-] generation

Production of OSCN[-] was assessed using the photometric 5-thio-2-nitrobenzoic acid (TNB) oxidation assay [24]. OSCN[-] converts TNB that absorbs light at 412 nm, into a colorless disulfide (5,5'-dithio-bis-[2-nitrobenzoic acid]) (DNTB, Ellman's reagent). OSCN[-] production is measured as decrease in OD at 412 nm and is calculated based on the Lambert-Beer Law and the absorption coefficient $\varepsilon_{412}$ = 14,100 $M^{-1}$ $cm^{-1}$ [25]. OSCN[-] production is expressed as concentration of OSCN[-] produced in the volume of the cell-free system under different conditions in 30 minutes.

### Statistical analysis

Significance among multiple samples was calculated using One-way or Two-Way ANOVA followed by Tukey's or Sidak's multiple comparison post-hoc tests. Significance between two samples was calculated using Mann-Whitney's test. Statistical analysis was performed using Prism 6 for Windows version 6.07 software. *, $p < 0.05$; **, $p < 0.01$; ***, $p < 0.001$.

## Results

### LPO-derived hypothiocyanite kills a diverse array of Spn strains

To explore the effects of OSCN[-] on *Spn*, we used our previously established cell-free *in vitro* system that generates OSCN[-] at levels comparable to those measured in human airways [26]. This system utilizes the enzyme glucose oxidase, which oxidizes its substrate glucose to produce D-gluconolactate [27]. $H_2O_2$ is a byproduct of the reaction and allows us to mimic the nature of $H_2O_2$ production *in vivo* by Dual oxidases [8, 26]. Previously, we have successfully used this experimental system to show that LPO-generated OSCN[-] inactivates a wide range of influenza strains [13, 27]. Using this $H_2O_2$/LPO/SCN[-] cell-free system, we wanted to determine the effectiveness of OSCN[-] against *Spn*. Physiologically relevant levels of each component of the system were utilized: 400 μM SCN[-] [28] and 6.5 μg/ml LPO [11]. $H_2O_2$ production by glucose oxidase was set to a rate of 0.01 U/ml, which is similar to what is seen in primary normal human bronchial epithelial (NHBE) cells by Duox [13, 29]. We tested both encapsulated (TIGR4, EF3030) and nonencapsulated (MNZ41) *Spn* strains, and we observed that the cell-free $H_2O_2$/LPO/SCN[-] system effectively killed all three strains of *Spn* (Fig 1A). Fig 1B shows that OSCN[-] is only produced when all components of the cell-free system are added. OSCN[-] is generated reproducibly to achieve a final OSCN[-] concentration of 41.2 ± 4.2 μM (mean ± S.E.M., n = 4) (Fig 1B). Since $H_2O_2$ alone is capable of killing *Spn* [30], we exposed EF3030 *Spn* to the same levels of glucose oxidase without SCN[-] or LPO to ensure our results were OSCN[-]-mediated. The results show that *Spn* survival is not impaired by $H_2O_2$ alone (Fig 1C).

### Catalase prevents the antimicrobial action of OSCN[-] on Spn

To further confirm our findings that OSCN[-] is antimicrobial against *Spn*, we utilized a kinetic assay to measure bacterial growth in presence of an inhibitor of the $H_2O_2$/LPO/SCN[-] cell-free system. Catalase is an enzyme found in almost all living organisms that catalyzes the decomposition of $H_2O_2$ to water and oxygen [31]. The use of catalase eliminates $H_2O_2$ and thereby OSCN[-] in our cell-free system (Fig 2A), rendering the cell-free system nonfunctional while

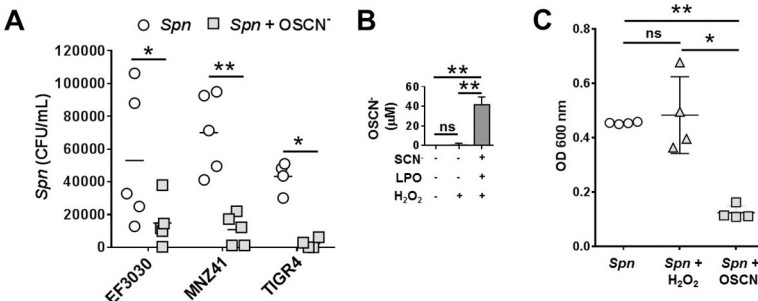

**Fig 1. LPO-derived hypothiocyanite kills Spn *in vitro*. (A)** OSCN⁻ mediated *Spn* killing was tested in the cell-free system against three different bacterial strains: EF3030 encapsulated serotype 19F (n = 5), MNZ41 nonencapsulated (n = 5) and TIGR4 encapsulated serotype 4 (n = 4). Bacteria were incubated for 6 hrs with or without OSCN⁻ generated by the LPO/SCN⁻/H₂O₂ system and bacterial killing was quantified by colony counts on BAP. Each dot represents a separate, individual experiment and their mean is also shown, Mann-Whitney test. **(B)** OSCN⁻ is only produced in the cell-free system when all of its components (LPO, SCN⁻, H₂O₂) are present. H₂O₂ is provided by the enzymatic reaction of glucose oxidase with glucose, not in a bolus-like fashion. Mean+/-S.E.M., n = 4, Two-way ANOVA and Sidak's multiple comparisons test. **(C)** *Spn* EF3030 growth is only inhibited by OSCN⁻ and not by H₂O₂. OSCN⁻ was generated in presence of the complete cell-free system (LPO+SCN⁻+glucose/GO) while H₂O₂ was produced by the glucose/GO system in the absence of LPO and SCN⁻. Bacterial growth was assessed using the microplate-based growth assay (n = 4). Each symbol represents a separate, individual experiment and their mean+/- S.E.M. is also shown, two-way ANOVA and Sidak's multiple comparisons test. Each experiment was done in biological triplicates and each biological replicate was tested in technical triplicates. (Ns, not significant; *, p<0.05; **, p<0.01. CFU, colony-forming unit; LPO, lactoperoxidase; OSCN⁻, hypothiocyanite; SCN⁻, thiocyanate; *Spn, Streptococcus pneumoniae*.

also ensuring that neither SCN⁻ nor LPO have an antimicrobial effect alone, independent of OSCN⁻ formation. Results shown in Fig 2B indicate that *Spn* bacteria exposed to OSCN⁻ have inhibited bacterial growth compared to those treated with catalase or unexposed to OSCN⁻. Both *Spn* strains tested show the same trend, where OSCN⁻ treatment significantly reduces bacterial growth (p<0.0001) and addition of catalase entirely rescues this effect (p<0.0001)

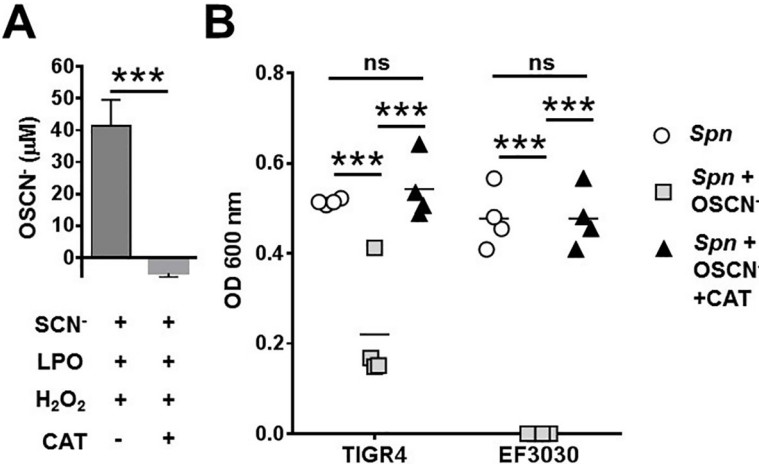

**Fig 2. Catalase rescues *Spn* from OSCN⁻ mediated growth inhibition. (A)** Addition of catalase to the cell-free system blocks OSCN⁻ generation measured by the DTNB assay. Error bars represent standard errors of the means, n = 4. Mann-Whitney test. **(B)** Catalase inhibits the bacteriostatic effect of OSCN⁻ on TIGR4 and EF3030 *Spn* strains (n = 4). Each symbol represents a separate, individual experiment and their mean is also shown, Two-way ANOVA, Tukey's multiple comparison test. Each experiment was done in biological triplicates and each biological replicate was tested in technical triplicates. Ns, not significant; *, p<0.05; **, p<0.01; ***, p<0.001. CAT, catalase.

(Fig 2B). The nonencapsulated *Spn* strain, MNZ41, could not be tested in this assay because it did not grow in liquid cultures used under these experimental conditions. Taken together, these data show for the first time in an *in vitro* model that OSCN$^-$ has a catalase-sensitive, antimicrobial action against different strains of *Spn*.

## LPO-derived OSCN$^-$ inhibits Spn growth in a SCN$^-$ concentration-dependent manner

Physiological levels of SCN$^-$ in the airways have been measured to be around 400 μM [28]. We decided to test SCN$^-$ in a supra- and superphysiological concentration range between 40 μM-4 mM to determine if killing of one of the *Spn* strains, EF3030, can be enhanced. Our data show that supraphysiological levels (40 μM) of SCN$^-$ kill *Spn* EF3030 in a robust manner (Fig 3 and 3B). Increasing the SCN$^-$ concentration showed a dose-dependent response, where bacterial killing continued to increase, all the way up to 4 mM of SCN$^-$ (Fig 3). From this data we conclude that the reported physiological SCN$^-$ concentration is sufficient to support the antibacterial activity of the LPO system against *Spn*.

## Commercially available 1$^{st}$ line™ effectively kills Spn via OSCN$^-$

The H$_2$O$_2$/LPO/SCN$^-$ cell-free system has proven to be effective at killing *Spn in vitro*. A drawback of this system is, however, that OSCN$^-$ has a very short life span (less than 30 minutes after it has been generated) requiring it to constantly be produced *in vitro* in order to test its effect on microbes. This is why we utilized glucose oxidase, not bolus-like addition of H$_2$O$_2$, to allow a steady production of H$_2$O$_2$ and OSCN$^-$ [27]. As the next step, we took advantage of and tested a commercially available product that also generates OSCN$^-$ and claims to keep it stable for much longer. This product, *1$^{st}$ line*™, utilizes a stabilizing molecule to allow OSCN$^-$ to persist for over 12 hours. We compared the bacteriostatic effect of the *1$^{st}$ line*™ product on *Spn* EF3030 and TIGR4. The results demonstrate that OSCN$^-$ generated by 1st line™ resulted in robust inhibition of *Spn* growth (Fig 4 and 4B). The addition of catalase during the generation of OSCN$^-$ with *1$^{st}$ line*™ also inhibited the antimicrobial action of OSCN$^-$ (Fig 4), similar to what was previously shown in our cell-free system (Fig 2). These results provide further evidence that OSCN$^-$ is solely responsible for the inhibition of *Spn* growth.

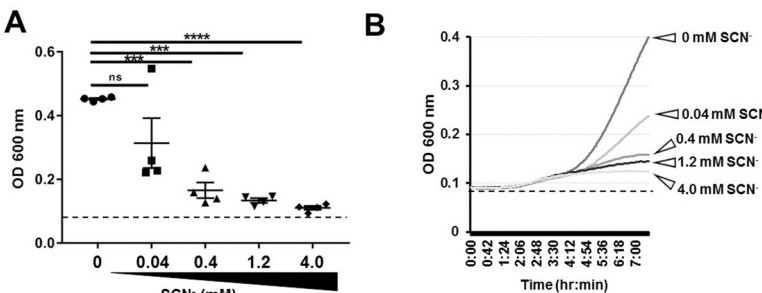

**Fig 3. *Spn* growth is inhibited in a thiocyanate dose-dependent manner. (A)** Increasing the concentration of SCN-in the cell-free system leads to improved antimicrobial effect of OSCN$^-$ against *Spn* EF3030. Bacteria were incubated for 6 hrs with LPO, glucose oxidase, glucose and different concentrations of SCN$^-$, and bacterial growth was followed by the microplate-based growth assay (n = 4). Each symbol represents a separate, individual experiment, mean+/-S.E. M. One-way ANOVA and Tukey's multiple comparison test. **(B)** Representative kinetics of EF3030 growth curves. Each experiment was done in biological triplicates and each biological replicate was tested in technical triplicates. ***, p<0.001; ns, not significant. Dotted lines indicate the OD background of the growth medium without bacteria that was not subtracted in these experiments.

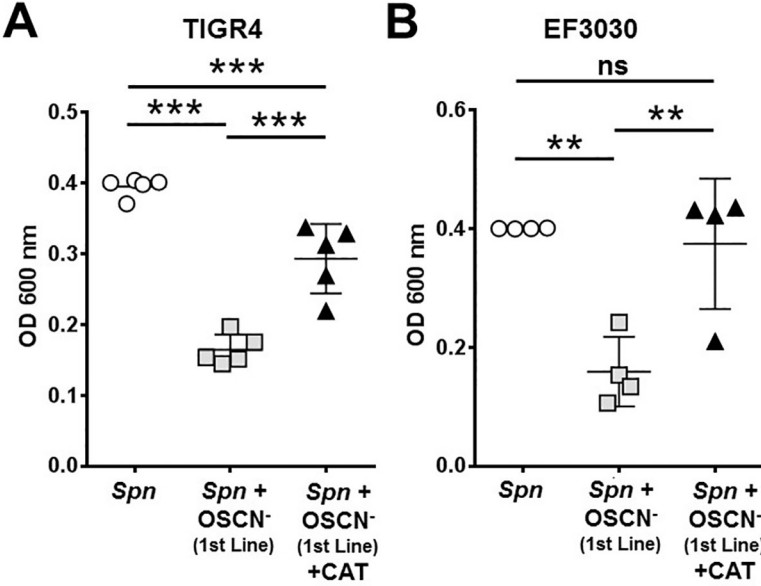

**Fig 4. OSCN⁻ generated using commercially available 1st line™ effectively inhibits *Spn* growth.** The 1st line™ product efficiently inhibits *Spn* growth: **(A)** TIGR4 (n = 5) and **(B)** EF3030 (n = 4) bacterial strains. Catalase reversed this effect partially (TIGR4) or fully (EF3030). Bacterial growth was measured by the microplate-based growth assay. Each symbol represents a separate, individual experiment, mean+/-S.E.M. One-way ANOVA, Tukey's multiple comparison test. Each experiment was done in biological triplicates and each biological replicate was tested in technical triplicates. **, p<0.01; ***, p<0.001; ns, not significant. CAT, catalase.

## The bacterial capsule is not protective against OCSN⁻

The bacterial capsule provides protection for *Spn* against threats of the environment including attacks by the immune system [32, 33]. We next asked whether the capsule provides protection for *Spn* against OSCN⁻. To answer this question, we exposed isogenic, capsule-deficient strains of *Spn* D39, EF3030 and TIGR4, and their corresponding, encapsulated, parental wild-type counterparts to OSCN⁻ in the cell-free experimental system and measured bacterial killing by CFU counting. Results in Fig 5 show that the capsule-free mutants were also susceptible to OSCN⁻ on all three backgrounds, similar to their encapsulated control strains. Catalase was also partially or fully effective in preventing the bactericidal effect of OSCN⁻ against all strains tested (Fig 5). Therefore, we conclude that the capsule provides no protection against the anti-pneumococcal action of OSCN⁻.

## The mechanism of action of OSCN⁻ is independent of lytA-mediated autolysis

Autolysis is a form of programmed cell death in bacteria including *Spn* that plays a roles in genetic exchange between bacterial cells, in eliminating damaged cells and has also been implicated in mediating the effects of antibiotics [34]. The *lytA* genes encodes a major autolysin (N-acetylmuramoyl-l-alanine amidase) in *Spn* that is a cell wall-degrading enzyme located in the cell envelope [35, 36]. Based on these, we postulated that *lytA*-mediated autolysis could represent the anti-pneumococcal mechanism of action of OSCN⁻. To test this hypothesis, we exposed *lytA*-competent and l*ytA*-deficient D39 *Spn* strains to OSCN⁻ in the cell-free system generated by 1st line™ and measured bacterial survival by colony counting. As the results in Fig 6 show, 1st line™ not only inhibits bacterial growth of *Spn* but also kills this bacterium in an

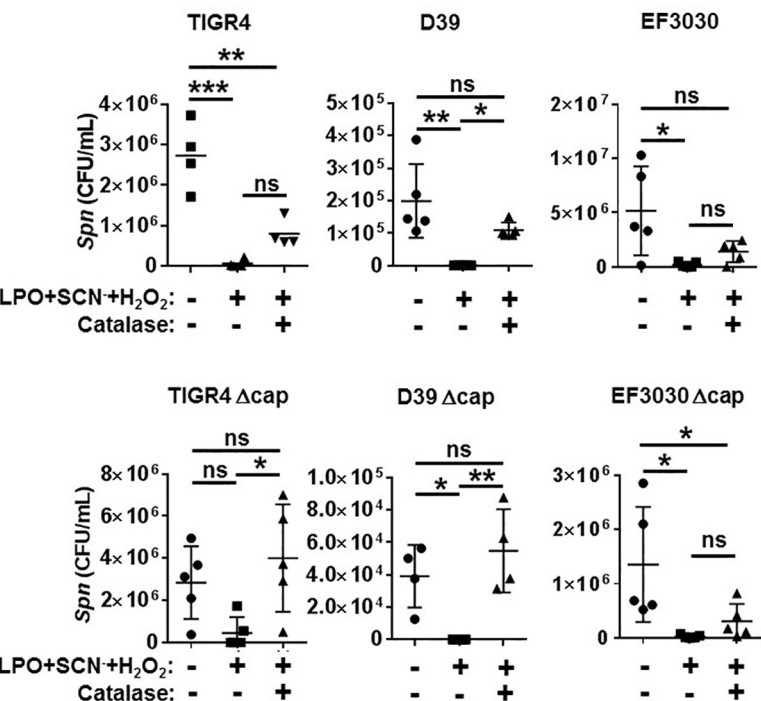

**Fig 5. LPO-derived hypothiocyanite kills both encapsulated and nonencapsulated *Spn* strains.** OSCN⁻mediated killing of *Spn* was tested in the cell-free system using 1st line ™ against three encapsulated strains of *Spn* (TIGR4, D39 and EF3030) and their capsule-deficient, isogenic mutants (Δcap) in 4–5 independent experiments: n = 4 for TIGR4 and D39 Δcap, n = 5 for the other strains. Bacteria were incubated for 6 hrs with or without OSCN⁻ generated by the LPO/SCN⁻/H₂O₂ system in presence or absence of catalase and bacterial killing was quantified by colony counts on BAP. Each symbol represents a separate, individual experiment, mean+/-S.E.M. is shown. ANOVA and Holm-Sidak's multiple comparisons test. Each experiment was done in biological triplicates and each biological replicate was tested in technical triplicates. Ns, not significant; *, p<0.05; **, p<0.01, ***, p<0.001. CFU, colony-forming unit; LPO, lactoperoxidase; OSCN⁻, hypothiocyanite; SCN⁻, thiocyanate; *Spn*, *Streptococcus pneumonia*; CAT, catalase.

OSCN⁻dependent manner. Fig 6 also shows that the *lytA*-deficient mutant was also killed efficiently by OSCN⁻ indicating that OSCN⁻ does not initiate *lytA*-mediated autolysis in *Spn*.

## Spn-derived H₂O₂ provides some protection against OSCN⁻mediated killing

Interestingly, not only human cells but *Spn* itself is capable of producing H₂O₂ [37]. *Spn*-generated H₂O₂ could interfere with the killing effect of OSCN⁻. *Spn*-generated H₂O₂ could provide additional H₂O₂ for OSCN⁻ generation by LPO leading to improved bacterial killing or it could prime bacteria against oxidative stress resulting in impaired *Spn* killing by OSCN⁻. To explore these possibilities, we tested a mutant TIGR4 *Spn* strain (Δ*spxB*) deficient in pyruvate oxidase, the main H₂O₂ producer in *Spn* [21], for susceptibility to OSCN⁻. As expected, the *spxB*-deficient TIGR4 strain had an H₂O₂ generation that was reduced by 72.5±0.9% (mean ± SEM, n = 2) compared to the wild-type TIGR4 counterpart (Fig 7A). The TIGR4Δ*spxB* mutant and its parental strain were exposed to OCSN⁻ produced by 1st line™, and bacterial killing and growth were evaluated with both, CFU-based counting and microplate-based bacterial growth assays. The TIGR4Δ*spxB* mutant was also susceptible to the antimicrobial effect of OSCN⁻ that was partially inhibited by catalase (Fig 7B). To better present the antibacterial action of OSCN⁻, we next defined and calculated "susceptibility to OSCN⁻" as the decrease in log₁₀ of the colony counts. CFU count results obtained at 6 hours of incubation reached the level of

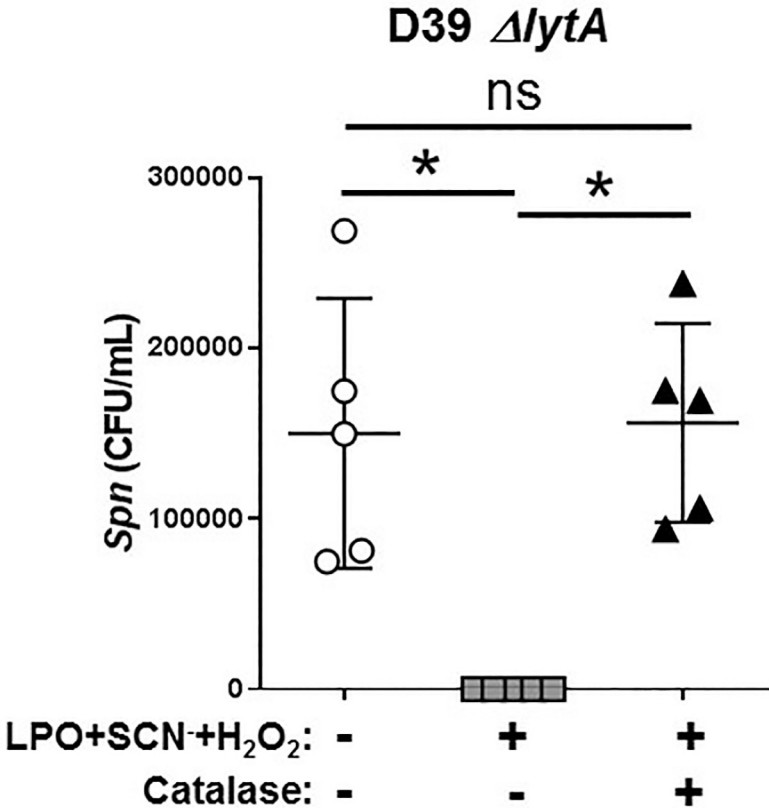

**Fig 6. LytA-deficiency does not protect Spn against OSCN⁻—mediated killing.** OSCN⁻—mediated killing of the lytA-deficient D39 *Spn* strain (*ΔlytA*) was tested in the cell-free system (n = 5). Bacteria were incubated for 6 hrs with or without OSCN⁻ generated by the 1st line™ and bacterial killing was quantified by colony counts on BAP. Each symbol represents a separate, individual experiment, mean+/-S.E.M. is shown. Mann-Whitney test. Each experiment was done in biological triplicates and each biological replicate was tested in technical triplicates. Ns, not significant; *, p<0.05. CFU, colony-forming unit; LPO, lactoperoxidase; SCN⁻, thiocyanate; *Spn, Streptococcus pneumoniae.*

significance (p = 0.029) (Fig 7C). Overall, we conclude that pyruvate oxidase provides improvement in *Spn* survival following OSCN⁻ exposure.

## Discussion

While the LPO-based system has been shown to be effective at killing numerous species of bacteria and viruses *in vitro*, no report to date studied the interaction between this system and *Spn. Spn* first encounters epithelial cells at the apical surface of the nasal cavity, a region that is a hotbed of defense mechanisms. The LPO-based system kills microbes in the extracellular space before they enter the epithelium and establish infection [8]. We were able to demonstrate that OSCN⁻ kills *Spn* effectively. The ability of OSCN⁻ to kill a variety of microbes in the extracellular spaces makes it a very interesting innate immune mechanism to study.

It is possible that *Spn* stimulates one of the main cellular sources of $H_2O_2$, Duox1, since we had previously shown that bacterial ligands of *P. aeruginosa* participate in Duox1 activation [12]. *Spn* has been shown to trigger $H_2O_2$ production in airway epithelial cells in a pneumolysin-independent but lytA-dependent manner [38]. We utilized the enzyme catalase, that converts $H_2O_2$ into $H_2O$ and $O_2$, to inhibit the production of OSCN⁻. This was necessary because *Spn* is a catalase-negative bacterium and is capable of producing its own $H_2O_2$ [39]. *Spn*-derived $H_2O_2$ is sufficient to mediate bactericidal activity of other bacteria and to stimulate

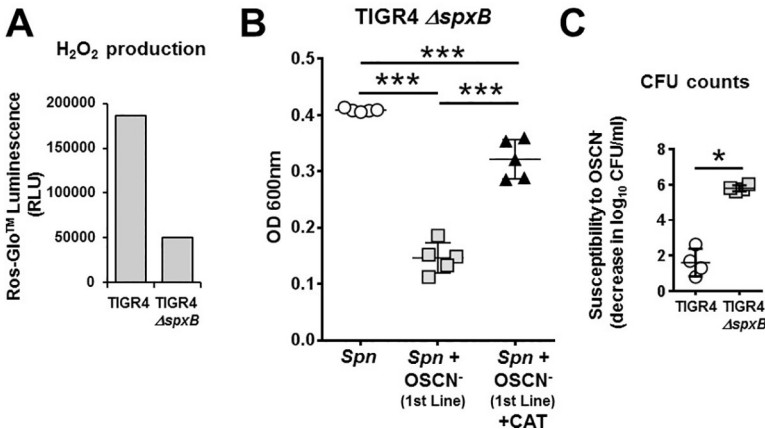

**Fig 7. Pyruvate oxidase-deficiency increases susceptibility of *Spn* to OSCN⁻. (A)** $H_2O_2$ generation was quantitated in wild-type and *spxB*-deficient (Δ*spxB*) TIGR4 *Spn* strains using the Ros-GLo™ luminescence $H_2O_2$ quantitation kit. Mean, n = 2. **(B)** OSCN⁻mediated (1st line™) killing of the TIGR4 Δ*spxB Spn* strain was measured by the microplate-based growth assay in presence or absence of catalase (n = 5). Each symbol represents a separate, individual experiment, mean+/-S.E.M. is shown. One-way ANOVA, Tukey's multiple comparison test. **(C)** Killing by OSCN⁻ generated via 1st line™ of wild-type and TIGR4 Δ*spxB Spn* strains was compared at 6 hrs via CFU counting (n = 4) and "susceptibility to OSCN⁻"was calculated as the decrease in $\log_{10}$ CFU upon exposure to OSCN⁻. Each symbol represents a separate, individual experiment, mean+/-S.E.M. is shown. Mann-Whitney's test. Each experiment was done in biological triplicates and each biological replicate was tested in technical triplicates. ***, p<0.001; *, p<0.05; ns, not significant. RLU, relative luminescence unit; CAT, catalase.

DNA damage and apoptosis in epithelial cells leading to tissue damage during infection [40]. It was found that in the absence of common antioxidant proteins, *Spn* utilizes pyruvate oxidase (SpxB) that has a dual role of creating $H_2O_2$, and protecting itself from oxidative damage [41]. Pyruvate oxidase, the main $H_2O_2$ source in *Spn* [21], has been found to be important to initiate oxidative attacks on host cells [40]. Our data indicate that SpxB provides a moderate protection against the oxidative attack of OCSN⁻. Our results are in line with previous findings of other groups where SpxB was determined to be useful against oxidative stress of different origins [41–43]. $H_2O_2$ generation by SpxB in *Spn* likely enhances its antioxidant capacity and thereby improves its defense against oxidative stress. Our results indicate a new, protective consequence of SpxB expression in *Spn*. The *in vivo* relevance of this finding remains to be confirmed in animal models.

*Spn* utilizes a wide spectrum of virulence factors, one of the most important ones being the polysaccharide capsule that forms the outermost layer of the bacteria [44]. This capsule provides protection against phagocytosis, complement components, mucus and spontaneous or antibiotic-induced autolysis [45]. Some *Spn* strains, however, do not possess a capsule [46, 47]. Our results show that both encapsulated and nonencapsulated *Spn* strains are susceptible to OSCN⁻. Thus, it is likely that OSCN⁻ is able to penetrate the capsule and interact with the cell wall or other internal bacterial components. Interestingly, catalase rescued OSCN⁻mediated *Spn* killing more efficiently when the capsule was absent, in case of TIGR4 and D39 (Fig 5). This difference seems to be strain-dependent as this effect was less pronounced in case of EF3030 (Fig 5). While the reason for this remains unclear, it could be related to differences in $H_2O_2$ generation or in inhibition of catalase by the capsule or other microbial factors among tested strains.

OSCN⁻ has a wide microbial target spectrum [8]. Its mechanism of action likely involves oxidative attack on one or more microbe-specific molecules or cellular mechanisms that essentially will lead to a bactericidal action. Previous studies have shown that OSCN⁻ is oxidizing

bacterial sulfhydryls [48–50], thereby inhibiting bacterial respiration [51], but no further research has been published to support this possible mechanism of action.

The LPO-based system presents an interesting therapeutic target, since it is an effective antimicrobial innate mechanism that does not have many drawbacks due to its final product being nontoxic to host cells and its broad activity against a wide range of pathogens [52]. By testing the $1^{st}$ line™ product (alongside our cell-free method of OSCN⁻ generation), we observed the same efficient *Spn* killing results. While we experienced similar levels of killing of *Spn* when comparing our glucose oxidase system and the $1^{st}$ line™ product, there are some subtle differences that would likely affect the efficacy of *in vivo* studies. The greatest benefit to the $1^{st}$ line™ system is the stabilizing aspect of the compound. Allowing OSCN⁻ to persist for over 12 hours would likely increase the efficacy and potency of the system which could be a major advantage over the non-stabilized, natural OSCN⁻ anion. This also allows for higher concentrations of OSCN⁻ to be achieved without having toxicity issues.

While these are encouraging, the study presented here has technical limitations as it was conducted solely in an *in vitro*, cell-free experimental system. *In vivo* conditions are obviously much more complex and could represent new challenges to prove the antibacterial efficacy of OSCN⁻. Duox1-dependent $H_2O_2$ production is regulated by several inflammatory and microbial stimuli *in vivo* and could be targeted by pneumococcal host evasion mechanisms not explored here. The *in vivo* life span of OSCN⁻ might be influenced by several additional biological factors that could not be tested in the *in vitro* system studied here. The *in vivo* administration route and formulation to boost airway production of OSCN⁻ offers numerous possibilities. Overall, these new questions will be answered in future studies using airway epithelial cultures, animal models and human patients.

We determined the effectiveness of OSCN⁻ against *Spn* in this proof-of-concept study as it had not been reported previously. We were able to demonstrate that OSCN⁻ effectively kills both encapsulated and nonencapsulated strains of *Spn* in a cell-free system. We also successfully utilized a commercially available product, $1^{st}$ line™, to demonstrate the therapeutic potential of the LPO system in an *in vitro* model. The mechanism of anti-pneumococcal action of OSCN⁻ is independent of the bacterial capsule or *lytA*-mediated autolysis but is opposed by the bacterial oxidant generator SpxB. These studies warrant further research to elucidate the molecular mechanism of antibacterial action of OSCN⁻.

## Author Contributions

**Conceptualization:** Aaron D. Gingerich, Balázs Rada.

**Data curation:** Aaron D. Gingerich, Fayhaa Doja, Rachel Thomason, Eszter Tóth, Martin V. Douglass, Balázs Rada.

**Formal analysis:** Aaron D. Gingerich, Fayhaa Doja, Rachel Thomason, Eszter Tóth, Martin V. Douglass, Larry S. McDaniel.

**Funding acquisition:** Balázs Rada.

**Investigation:** Balázs Rada.

**Methodology:** Aaron D. Gingerich, Rachel Thomason, Jessica L. Bradshaw, Larry S. McDaniel, Balázs Rada.

**Project administration:** Larry S. McDaniel, Balázs Rada.

**Resources:** Larry S. McDaniel, Balázs Rada.

**Supervision:** Balázs Rada.

**Validation:** Jessica L. Bradshaw, Balázs Rada.

**Writing – original draft:** Aaron D. Gingerich.

**Writing – review & editing:** Aaron D. Gingerich, Jessica L. Bradshaw, Larry S. McDaniel, Balázs Rada.

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
