## [Decision Letter · Decision Letter 0]

18 Sep 2019

PONE-D-19-23735

Oxidative killing of encapsulated and nonencapsulated Streptococcus pneumoniae by lactoperoxidase-generated hypothiocyanite

PLOS ONE

Dear Dr. Rada,

Thank you for submitting your manuscript to PLOS ONE. After careful consideration, we have decided that your manuscript does not meet our criteria for publication and must therefore be rejected.

Specifically:

In addition to reviewer 1, I have myself reviewed the manuscript. Although reviewer 1 comments read a bit harsh at times, I have to agree that overall the presented results are merely confirmative of numerous publications that have shown effectiveness of the LPO-isothiocyanate system against streptococci that inhabit the oral cavity and respiratory system. 

It would have been more informative if the effectiveness was investigated under more relevant biofilm conditions or in a multispecies setting. Mimicking the mucus environment of the respiratory system could have provided novel insights into killing kinetics. For example, would slow growth and limited diffusion affect the antimicrobial activity?

Further, it is not clear what growth conditions were used for some of the experiments. For example, the growth curve in Fig. 3B, what was the growth medium?

Also, no effect of SpxB produced hydrogen peroxide was observed. Could that be simply due to the fact that under the here used conditions, SpxB is not active?

I am sorry that we cannot be more positive on this occasion, but hope that you appreciate the reasons for this decision.

Yours sincerely,

Jens Kreth

Academic Editor

PLOS ONE

Reviewers' comments:

Reviewer's Responses to Questions

**Comments to the Author**

1. Is the manuscript technically sound, and do the data support the conclusions?

Reviewer #1: No

2. Has the statistical analysis been performed appropriately and rigorously? 

Reviewer #1: I Don't Know

3. Have the authors made all data underlying the findings in their manuscript fully available?

Reviewer #1: No

4. Is the manuscript presented in an intelligible fashion and written in standard English?

Reviewer #1: No

5. Review Comments to the Author

Reviewer #1: When one steps away from the hyperbole …

How can the LPO system (LPO + H2O2 + SCN- = HOSCN) be a “novel, anti-pneumococcal therapy” when it is a principal component in the innate defense stratagem of the endocrine system, and there already exists an abundance of literature (100 + papers?) that evidence the effectiveness of the LPO system against streptococci, dating back more than 70 years.

the present paper is at best pedestrian. The claim that this is the first study of the efficacy of the LPO system vs. Streptococcus pneumoniae is false. Use the Google machine.

Comically, the authors test a undefined "commercial product" by Profound Products that “stabilizes” HOSCN. According to the description the “1st Line” product, it also contains as “other” ingredients “hydrogen peroxide, poly aluminum chloride, lactoperoxidase and bentonite (note that no hydrogen peroxide or aluminum is consumed as these are converted by the enzyme in the manufacture of thiocynate (sp) ions).” We note that thiocyanate is not produced by these ingredients, and LPO is inactivated by hydrogen peroxide in the absence of thiocyanate. This reviewer has knowledge this product actually contains iodide, and HOSCN oxidizes iodide to yield iodine, the probable active ingredient.

l.128: The LPO system described effectively generates a bolus of HOSCN. It is not clear if the HOSCN was generated exogenously, then added to the wells.

l.133: It is not at all clear what grown medium was used. Was the medium just HBSS or BAP? If HBSS, this stressor would not realistically reflect the infectious agent in vivo. HOSCN reacts instantly with BAP.

l.193: “… short-lived TNB cannot be purchased and was generated by reducing DTNB with the help of beta-mercaptoethanol”. First, TNB is not short-lived. We have a flask of it that was synthesized more than a decade ago and shows no sign of decomposition. Second, if TNB was synthesized from DTNB in situ using ME, there is a likelihood that unreactive ME remains, which would render the assay inaccurate.

l.237: “H2O2 is provided by the enzymatic reaction of glucose oxidase with glucose, not in a bolus-like fashion.” … the authors clearly do not understand enzyme kinetics. I calculate a half-life of about 18 seconds for this reaction under the conditions employed.

l.249: Yes, catalase, the most efficient enzyme known to man, removes one of the essential components of the LPO system.

l. 278: Figure 3B exhibits a profound ignorance of the LPO system. Under the conditions described in the experimental section, only [SCN-] > 50 uM will produce HOSCN under their reaction conditions. Also, the assay does not test for regrowth (the conditions are likely inhibitory, not cytocital).

l.337: Been shown before.

l.401: HOSCN only attacks sulfhydryl groups.

l.407: So is bleach.

l.411: I am aware of at least two patents for the use of the LPO system in the treatment of lung infections.

I would like to know if the authors have received any financial support from Profound Products.

Do not publish … anywhere.

6. PLOS authors have the option to publish the peer review history of their article (what does this mean?). If published, this will include your full peer review and any attached files.

Reviewer #1: No

- - - - -

---

## [Author Response · Author response to Decision Letter 0]

12 Feb 2020

Response to the critiques

Responses to the editor's critiques:

We disagree that our findings are merely confirmative of previous findings. While several Streptococcus species have already been found to be sensitive to OSCN-, no published scientific literature is out there to show this for the clinically most important Streptococcus species, Streptococcus pneumoniae. I know this is surprising but we searched PubMed thoroughly and did not find any publication showing the effectiveness of the LPO-based system in any experimental setup against S. pneumoniae. We actually wrote the most up-to-date review in 2019 on the topic and listed all the 100+ publications studying several microbial species that have been studied for LPO-mediated killing, but S. pneumoniae is not there. We think our data is an important set of information completely suitable to your journal showing that the clinically most relevant Streptococcus species, S. pneumoniae, is sensitive to OCSN- in vitro. This is also important because we have unpublished data showing that some Streptococcus species are not sensitive, so one cannot simply assume that all Streptococcus species are sensitive to OSCN-. 

Regarding the criticism of not characterizing the spxB-deficient Spn mutant, there are actually data shown in the manuscript that SpxB is active under our experimental conditions in Fig. 7A as the spxB-deficient mutant had about 80% reduction in H2O2 production compared to the WT strain. The killing experiments were done under the same experimental conditions.

The editor is right, we forgot to add what medium was used in the growth assay. Bacteria were grown on BAP medium. This was added to the revised manuscript. 

Response to the reviewer:

Critique: How can the LPO system (LPO + H2O2 + SCN- = HOSCN) be a “novel, anti-pneumococcal therapy” when it is a principal component in the innate defense stratagem of the endocrine system, and there already exists an abundance of literature (100 + papers?) that evidence the effectiveness of the LPO system against streptococci, dating back more than 70 years.

Response: As also detailed above in our answer to the editor, no one documented the effect of this system against S. pneumoniae (Pneumococcus) before. I agree that several other Streptococcal species were published but not S. pneumoniae. We cited them all in our review published in 2019 in the Journal of Microbiology (https://www.ncbi.nlm.nih.gov/pubmed/29858825). So, based on this, using/targeting the OSCN-based mechanism provides theoretically a novel strategy to kill pneumococcus, not other Streptococci. Several mechanisms of the immune system can be and have been targeted to improve bacterial clearance, so the same can be done with the LPO-based system. At the same time, we acknowledge that the results are restricted to an in vitro system.Therapeutic considerations might be overreaching; therefore, we removed this part of the discussion from the revised version. 

The reviewer’s comment related to the endocrine system is unclear? We never said the LPO-based system is an endocrine system.

Critique: the present paper is at best pedestrian. The claim that this is the first study of the efficacy of the LPO system vs. Streptococcus pneumoniae is false. Use the Google machine.

Response: We used both the Google and PubMed “machines” and there is no publication to show that OSCN kills Streptococcus pneumoniae. The reviewer is more than welcome to refer to papers showing Spn killing by OSCN in the literature.

Critique: Comically, the authors test a undefined "commercial product" by Profound Products that “stabilizes” HOSCN. According to the description the “1st Line” product, it also contains as “other” ingredients “hydrogen peroxide, poly aluminum chloride, lactoperoxidase and bentonite (note that no hydrogen peroxide or aluminum is consumed as these are converted by the enzyme in the manufacture of thiocynate (sp) ions).” We note that thiocyanate is not produced by these ingredients, and LPO is inactivated by hydrogen peroxide in the absence of thiocyanate. This reviewer has knowledge this product actually contains iodide, and HOSCN oxidizes iodide to yield iodine, the probable active ingredient.

Response: We defined the product clearly in the methods section and only referred to it in the text as commercial product. We can provide any additional information, as much as needed. Of course, the 1st Line product has LPO, H2O2, SCN-, polyaluminum chloride and bentonite in it –as they state in their description. Why would SCN- be produced by these ingredients when it is provided as one of the substrates? This comment was unclear. We measured OSCN- production with this kit and found it to be comparable to the levels obtained by the glucose/glucose oxidase system. SCN- is provided in the kit, so LPO is unlikely to be inactivated by hydrogen peroxide in the absence of thiocyanate – as the reviewer stated it. We cannot comment on the undocumented knowledge the reviewer claims to have on this product. No data are shown or referred to. 

Critique: l.128: The LPO system described effectively generates a bolus of HOSCN. It is not clear if the HOSCN was generated exogenously, then added to the wells.

Response: We might have different definitions for “bolus”. When we used the term “bolus”, the compound was not produced by enzymatic reaction but provided already synthesized in an unnaturally high concentration in one moment. HOSCN was generated in the wells, as described in the methods.

Critique: l.133: It is not at all clear what grown medium was used. Was the medium just HBSS or BAP? If HBSS, this stressor would not realistically reflect the infectious agent in vivo. HOSCN reacts instantly with BAP.

Response: HBSS was used in the killing assay as stated in the methods and bacteria were then inoculated on BAP for growth. This experiment uses an in vitro system with its obvious limitations and advantages as a model of the in vivo environment. Similar buffers were used in most of the published papers using such in vitro systems to study the antimicrobial effect of this LPO-based system. The information gathered is useful to document that another bacterial species, S. pneumoniae is also killed by the LPO/SCN- system. How much is this true and reflected in vivo, remains to be clarified not only for S. pneumoniae, but for all the other pathogens published since –paradoxically- the relevance of this antimicrobial system has not yet been established in any mammalian organism including mice and humans. We added more information about the growth conditions of bacteria to the revised manuscript.

Critique: l.193: “… short-lived TNB cannot be purchased and was generated by reducing DTNB with the help of beta-mercaptoethanol”. First, TNB is not short-lived. We have a flask of it that was synthesized more than a decade ago and shows no sign of decomposition. Second, if TNB was synthesized from DTNB in situ using ME, there is a likelihood that unreactive ME remains, which would render the assay inaccurate.

Response: We read several publications that measured OSCN- production by this simple method and all claimed that TNB is not stable over long period of time. We, therefore, decided to purchase DTNB and reduce it with the help of ME, as dozens of related manuscripts did it too. 

Critique: l.237: “H2O2 is provided by the enzymatic reaction of glucose oxidase with glucose, not in a bolus-like fashion.” … the authors clearly do not understand enzyme kinetics. I calculate a half-life of about 18 seconds for this reaction under the conditions employed.

Response: We actually backed up our decision for using this method of H2O2 production with more meaningful, and for the studied topic, more useful hydrogen peroxide measurements. Adding 1 mM bolus H2O2 to the system -as is done by several investigators in the field- has nothing to do with the nature of H2O2 production by their physiological source, bronchial epithelial cells. We measured and optimized H2O2 production of the glucose/glucose oxidase system so that it mimics the slow but relatively steady nature and amplitude of H2O2 production by airway cells. While it is true that –of course- the rate of H2O2 production slowed over time, it was still detectable at 60 minutes. Despite its deceleration over time, it is still a better approach to model H2O2 generation by airway cells than pouring a bucket load of physiologically irrelevant concentration of H2O2 into our system.

Critique: l.249: Yes, catalase, the most efficient enzyme known to man, removes one of the essential components of the LPO system.

Response: While the effect of catalase is obvious, one cannot simply assume that it will also work in our experimental system and has to show results. Removing a critical component of a system is an essential approach to show how the system works and this approach should rather be appreciated and encouraged by the reviewer to show the H2O2-dependence of the studied mechanism.

Critique: l. 278: Figure 3B exhibits a profound ignorance of the LPO system. Under the conditions described in the experimental section, only [SCN-] > 50 uM will produce HOSCN under their reaction conditions. Also, the assay does not test for regrowth (the conditions are likely inhibitory, not cytocital).

Response: I do not see any ignorance of the LPO system here and in fact the reviewer’s calculations clearly confirm that OSCN- is responsible for the inhibitory effect of microbial growth observed in Fig 3B as the lowest SCN- concentration used is 40 uM. The second observation is not new, at all, as we clearly stated in the manuscript that the growth assay used in figure 3B measures inhibition of bacterial growth and not direct microbicidal activity. We had the CFU assay for that. 

Critique: l.337: Been shown before.

Response: No, it has not. Please cite any report from the medical literature and we will immediately retract our submission.

Critique: l.401: HOSCN only attacks sulfhydryl groups.

Response: That’s what we said too, so we are unclear about the point of this critique. In line 401 we referred to the long list of microbes OSCN- has been described to target, if that was confusing.

Critique: l.407: So is bleach.

Response: Natural products of the immune system have been and are being used as part of therapies for a long list of diseases.

Critique: .411: I am aware of at least two patents for the use of the LPO system in the treatment of lung infections.

Response: So are we. We cited them in our last year’s review in Journal of Microbiology. None of them, however, claims to be targeting S. pneumoniae. We refer to its novelty to be used against Pneumococcus, not other bacteria. 

Critique: I would like to know if the authors have received any financial support from Profound Products.

Response: No, we have not. We would have acknowledged it in the manuscript if we had according to the policies of PLoS One and any other journal. We used this product because this was the only, purchasable product we could find. We are glad to test any other, commercially available OSCN-producing product in the future.

---

## [Decision Letter · Decision Letter 1]

15 May 2020

PONE-D-19-23735R1

Oxidative killing of encapsulated and nonencapsulated Streptococcus pneumoniae by lactoperoxidase-generated hypothiocyanite

PLOS ONE

Dear Rada,

Thank you for submitting your manuscript to PLOS ONE. After careful consideration, we feel that it has merit but does not fully meet PLOS ONE’s publication criteria as it currently stands. Therefore, we invite you to submit a revised version of the manuscript that addresses the points raised during the review process, which mostly refer to changes in the text of your manuscript.

We would appreciate receiving your revised manuscript by Jun 29 2020 11:59PM. To enhance the reproducibility of your results, we recommend that if applicable you deposit your laboratory protocols in protocols.io, where a protocol can be assigned its own identifier (DOI) such that it can be cited independently in the future. For instructions see: http://journals.plos.org/plosone/s/submission-guidelines#loc-laboratory-protocols

We look forward to receiving your revised manuscript.

Kind regards,

Mariola J Edelmann, Ph.D. and Filippo Giarratana

Academic Editors

PLOS ONE

Journal Requirements:

1. Please ensure that your manuscript meets PLOS ONE’s style requirements, including those for file naming. The PLOS ONE style templates can be found at http://www.plosone.org/attachments/PLOSOne_formatting_sample_main_body.pdf and http://www.plosone.org/attachments/PLOSOne_formatting_sample_title_authors_affiliations.pdf

3. Thank you for updating your data availability statement. You note that your data are available within the Supporting Information files, but no such files have been included with your submission. At this time we ask that you please upload your minimal data set as a Supporting Information file, or to a public repository such as Figshare or Dryad.

Please also ensure that when you upload your file you include separate captions for your supplementary files at the end of your manuscript.

As soon as you confirm the location of the data underlying your findings, we will be able to proceed with the review of your submission.

Reviewers' comments:

Reviewer's Responses to Questions

**Comments to the Author**

1. If the authors have adequately addressed your comments raised in a previous round of review and you feel that this manuscript is now acceptable for publication, you may indicate that here to bypass the “Comments to the Author” section, enter your conflict of interest statement in the “Confidential to Editor” section, and submit your "Accept" recommendation.

Reviewer #2: All comments have been addressed

Reviewer #3: (No Response)

2. Is the manuscript technically sound, and do the data support the conclusions?

Reviewer #2: Yes

Reviewer #3: Yes

3. Has the statistical analysis been performed appropriately and rigorously? 

Reviewer #2: Yes

Reviewer #3: Yes

4. Have the authors made all data underlying the findings in their manuscript fully available?

Reviewer #2: Yes

Reviewer #3: No

5. Is the manuscript presented in an intelligible fashion and written in standard English?

Reviewer #2: Yes

Reviewer #3: Yes

6. Review Comments to the Author

Reviewer #2: In this manuscript Gingerich and colleagues test an lactoperoxidase(LPO)-based antibacterial system against S. pneumoniae. LPO is a natural antibacterial agent that plays an important role in the innate immune system. LPO has been shown to have some efficacy in treating certain oral diseases. However, as might be expected translation of strong in vitro data to strong activity in vivo is a challenge (because several components are required that do not retain optimal activity in human fluids). Nevertheless, there is some merit to studying the activity of LPO against clinically relevant bacteria. For this submission I am reviewing a previously reviewed version of the manuscript. The reviewer comments and responses were combative, predominately focused on novelty. I have mostly focused on the rigor of the studies, in line with the acceptance guidelines of PLOS ONE. The manuscript is well written, and the experiments are described in a logical manner. The methods are described in good detail and could be reproduced by others. The antibacterial system has dose-dependent efficacy and the authors study several components of S. pneumoniae physiology that could impact efficacy (capsule, autolysis and spxB). I have some minor comments as follows:

Line 409. Change ‘Spxb’ to ‘SpxB’

Discussion. It might be relevant to add a paragraph on limitations/technical challenges that will need to be addressed before the product can be used in vivo – these may well be future experiments that the authors would like to conduct. There is clear antibacterial efficacy in vitro but translation to in vivo efficacy will be a challenge, and something that most readers will contemplate.

Reviewer #3: Overall, this is a scientifically sound study and the conclusions are supported by the presented experiments. The authors have also adequately addressed the concerns raised by previous reviewers. In particular, this reviewer's own PubMed search failed to find any studies that documented the effect of OSCN- specifically against Spn. However, the following comments would help improve the clarity of the data presented in this manuscript:

1. For all figures, it is unclear if the reported data represents technical or biological replicates.

2. For all bacteria killing assays assessed by CFU, the methods section states that "A time zero condition was also counted to make sure that bacterial death was due to OSCN- and not related to an unknown variable". It is agreed that is an important control, but these time zero CFU counts do not appear to be reported in any of the related figures (1A, 5, 6, 7C). Please include this data.

3. In some instances, data derived from the bacteriostatic assay (where growth inhibition is measured by OD in a microtiter plate assay) is incorrectly referred to or discussed as demonstrating "killing", when in fact this assay is measuring growth inhibition. (ex: Figures 2 and 3 legend titles, results line 289).

4. For all figures reporting CFU data, converting the Y axis to log scale would improve resolution of the actual CFU data points that cluster around the "0" Y axis point (ex: when bacteria are incubated in the presence of OSCN-).

5. The data presented in Figure 5 suggest that in most cases, the catalase control worked better in the capsule mutants compared to their parental strains; is there a reason this might be the case?

6. For the experiment in Figure 7 (comparing wildtype and pyruvate oxidase mutant killing by OSCN-), why was the "1st line" product used rather than the in vitro assay used in Figure 1? The fact that the mutant was more susceptible to OSCN- is very interesting. Was this phenotype genetically complemented? Figure 7C is somewhat confusing, why not present the Y axis data as CFU counts?

7. Results line 339: the term "allolysis" (instead of autolysis) is more appropriate when discussing in this context.

7. PLOS authors have the option to publish the peer review history of their article (what does this mean?). If published, this will include your full peer review and any attached files.

Reviewer #2: Yes: Robert C. Shields

Reviewer #3: No

---

## [Author Response · Author response to Decision Letter 1]

18 Jun 2020

Response to the critiques

We thank the editor for allowing a resubmission and re-review of our revised manuscript. We also thank the reviewers for their constructive criticisms of our manuscript that have significantly improved it. We made every effort to address all the comments. We hope that the manuscript is now acceptable for publication.

- Response to the editor:

According to the editor’s request, we made sure that the manuscript meets PLOS ONE’s style requirements, including those for file naming. The single data set that was referred to as “data not shown” in the original, not the revised, version of the manuscript has been removed. No need, therefore, for supplementary data deposition. We are also attaching and uploading the minimal data set as a Supporting Information file requested by your journal.

- Responses to comments of reviewer 2:

Critique: Line 409. Change ‘Spxb’ to ‘SpxB’

Response: Thanks for pointing this out, we made the change in the revised text.

Critique: Discussion. It might be relevant to add a paragraph on limitations/technical challenges that will need to be addressed before the product can be used in vivo – these may well be future experiments that the authors would like to conduct. There is clear antibacterial efficacy in vitro but translation to in vivo efficacy will be a challenge, and something that most readers will contemplate.

Response: We thank the reviewer for bringing this important issue up. According to this, a separate paragraph has been added to the end of the discussion to mention all the limitations of our current study using a cell-free, in vitro experimental system.

- Responses to the comments of reviewer 3:

Critique: For all figures, it is unclear if the reported data represents technical or biological replicates.

Response: We thank the reviewer for this request. We added this information to all the figure legends in the revised manuscript.

Critique: For all bacteria killing assays assessed by CFU, the methods section states that "A time zero condition was also counted to make sure that bacterial death was due to OSCN- and not related to an unknown variable". It is agreed that is an important control, but these time zero CFU counts do not appear to be reported in any of the related figures (1A, 5, 6, 7C). Please include this data.

Response: We thank the reviewer for this comment. While we agree that this data serve as important controls, we consider them quality controls of the assay, therefore, decided not to report them individually at each figure but to expand the related sentence in the description of the CFU killing method to this:

“A time 0 condition was also counted to make sure that bacterial death was due to OSCN- and not related to an unknown variable, and that no significant changes in bacterial numbers were observed in samples containing only bacteria during the duration of the experiments.”

Critique: In some instances, data derived from the bacteriostatic assay (where growth inhibition is measured by OD in a microtiter plate assay) is incorrectly referred to or discussed as demonstrating "killing", when in fact this assay is measuring growth inhibition. (ex: Figures 2 and 3 legend titles, results line 289).

Response: We tried to be consistent with the related nomenclature of “killing” vs “growth inhibition” throughout the manuscript but obviously missed these few occasions. We thank the reviewer for identifying them. The terms “killing” or “killed” were changed in both instances to “growth inhibition”.

Critique: For all figures reporting CFU data, converting the Y axis to log scale would improve resolution of the actual CFU data points that cluster around the "0" Y axis point (ex: when bacteria are incubated in the presence of OSCN-).

Response: We had long discussions in the group whether the linear or log scale results should be presented in the figures with CFU data. We had decided to present the results on linear scale, instead of log scale, because we think it better shows the impressive antibacterial effect of OSCN- against Spn. 

Critique: The data presented in Figure 5 suggest that in most cases, the catalase control worked better in the capsule mutants compared to their parental strains; is there a reason this might be the case?

Response: We thank the reviewer for this interesting observation. Indeed, catalase worked better in case of TIGR4 and D39 in the absence of the capsule while this difference was less pronounced in case of EF3030 (Fig. 5). While we do not have any proven explanation for this at this point, we have added a few sentences about this to the discussion pointing to potential explanations. 

Critique: For the experiment in Figure 7 (comparing wild0type and pyruvate oxidase mutant killing by OSCN-), why was the "1st line" product used rather than the in vitro assay used in Figure 1? The fact that the mutant was more susceptible to OSCN- is very interesting. Was this phenotype genetically complemented? Figure 7C is somewhat confusing, why not present the Y axis data as CFU counts?

Response: In figure 6 and 7 we did not only want to test the effect of the respective bacterial mutants but also aimed at delivering the first results to show that “1st line” is not only bacteriostatic but also directly kills Spn. This was actually mistakenly stated in the figure 6 legend and now it has been corrected to “1st line”. A sentence was added about the direct Spn-killing effect of 1st line to the figure 6 section of the results. 

No, the SpxB-deficient mutant was not complemented. 

We would like to keep the “susceptibility to OSCN’” term as introduced in figure 7 as we think it better explains the antimicrobial action of OSCN- than showing the CFU results. We are aware of it that it might get somewhat confusing, therefore we added an extra explanation to the corresponding part of the results section.

 Critique: Results line 339: the term "allolysis" (instead of autolysis) is more appropriate when discussing in this context.

Response: We thank the reviewer for the comment. However, we feel in the overall context of the manuscript autolysis is the response we tested and chose not to change the wording. Classically, autolysis is associated with lytA expression and lysis of self, while allolysis is more a term of fratricide (lysis of nearby cells). Allolysis is also associated with the competence system and includes up to seven genes. In our study, we used a specific lytA gene knock out strain to look at the effects of autolysis. We did not look at the impact of differences in competence or fratricide in the in vitro model of Spn killing, so allolysis is not an appropriate term to be used based on our experimental approach.

---

## [Decision Letter · Decision Letter 2]

8 Jul 2020

Oxidative killing of encapsulated and nonencapsulated Streptococcus pneumoniae by lactoperoxidase-generated hypothiocyanite

PONE-D-19-23735R2

Dear Dr. Rada,

We’re pleased to inform you that your manuscript has been judged scientifically suitable for publication and will be formally accepted for publication once it meets all outstanding technical requirements.

Kind regards,

Mariola J Edelmann, Ph.D.

Academic Editor

PLOS ONE

Reviewers' comments:

Reviewer's Responses to Questions

**Comments to the Author**

1. If the authors have adequately addressed your comments raised in a previous round of review and you feel that this manuscript is now acceptable for publication, you may indicate that here to bypass the “Comments to the Author” section, enter your conflict of interest statement in the “Confidential to Editor” section, and submit your "Accept" recommendation.

Reviewer #2: All comments have been addressed

Reviewer #3: All comments have been addressed

2. Is the manuscript technically sound, and do the data support the conclusions?

Reviewer #2: Yes

Reviewer #3: (No Response)

3. Has the statistical analysis been performed appropriately and rigorously? 

Reviewer #2: Yes

Reviewer #3: (No Response)

4. Have the authors made all data underlying the findings in their manuscript fully available?

Reviewer #2: Yes

Reviewer #3: (No Response)

5. Is the manuscript presented in an intelligible fashion and written in standard English?

Reviewer #2: Yes

Reviewer #3: (No Response)

6. Review Comments to the Author

Reviewer #2: (No Response)

Reviewer #3: (No Response)

7. PLOS authors have the option to publish the peer review history of their article (what does this mean?). If published, this will include your full peer review and any attached files.

Reviewer #2: **Yes: **Robert Shields

Reviewer #3: No

---

## [Editor Report · Acceptance letter]

17 Jul 2020

PONE-D-19-23735R2 

Oxidative killing of encapsulated and nonencapsulated Streptococcus pneumoniae by lactoperoxidase-generated hypothiocyanite 

Dear Dr. Rada:

I'm pleased to inform you that your manuscript has been deemed suitable for publication in PLOS ONE. Congratulations! Your manuscript is now with our production department. 

Kind regards, 

on behalf of

Dr Mariola J Edelmann 

Academic Editor

PLOS ONE